# Unified Transformer Framework for Active Adaptation to Concept Drift in Time Series Forecasting

**Ekkanat Tanchavalit**[1]    **Alina Dambaeva**[1]    **Elvin Chen**[1]    **Azza Alawady**[1]
[1]Department of Computer Science and Technology, Tsinghua University
{chenyiha25, dln25, cpr25, anxy25}@mails.tsinghua.edu.cn

## 1 Introduction, Gaps, and Novelty

Modern timeseries forecasting models are built on the fragile assumption of stationarity, failing to account for the inherent non-stationarity of real-world data [19, 21]. While phenomena like trend and seasonality are well-documented, "concept drift," where the functional mapping between historical data and future values evolves, remains a critical bottleneck [20, 22]. Existing approaches often assume stationarity or deal with non-stationarity by differencing, decomposition [14], or normalization that fundamentally collapse when the underlying data-generating process shifts [7]. Alternatively, Test-time adaptation (TTA), either passive or active, is a promising paradigm where a pre-trained model adapts to test data in real time [5].

### 1.1 Gaps Identified

Current test-time adaptation (TTA) methods are either computationally prohibitive, requiring expensive optimization and ground-truth labels that are unavailable in real-time, or prone to catastrophic forgetting. Active adaptation strategies often rely on simplistic statistics that miss complex, multivariate shifts in input-output relationships. Also, while attention-based methods exist [6, 1], they often treat drift detection and model adaptation as disconnected components, failing to provide a unified architecture that can both identify and mitigate drift online.

### 1.2 Novelty

This project introduces a unified, transformer-based framework [11, 8] that couples concept drift detection directly to an adaptation mechanism within a single architecture. The core novelty lies in leveraging pre-softmax attention dynamics to capture directional changes in temporal and channel dependencies as an early warning for drift, building upon insights from recent attentive forecasting models [8, 9, 13]. To address the computational overhead of continuous learning, the framework employs Parameter-Efficient Fine-Tuning (PEFT), specifically Low-Rank Adaptation (LoRA) [4]. This allows for rapid, lightweight updates to newly detected regimes, ensuring the model evolves in real-time without the risk of catastrophic forgetting or the need for extensive memory storage.

## 2 Motivation

Concept drift degrades forecasting models in high-stakes settings. In human activity recognition (USC-HAD [18]), a model trained on seated desk work may fail when the same user walks outdoors—sensor distributions shift without ground-truth label to trigger retraining. In power grid management (ETT), gradual drift in transformer oil temperature relationships across seasons can cause errors in load forecasts by critical margins. On freeways (Traffic), an abrupt shift from normal flow to holiday or accident conditions invalidates occupancy predictions within minutes. In each case, there can be a substantial cost of failure. As environments grow more non-stationary, detecting and adapting to drift online without labels has become an urgent operational need.

## 3 Background

This proposal addresses online time series forecasting under concept drift. Unlike standard forecasting, no ground-truth labels are available during test time, and adaptation must occur incrementally as data streams in.

Several concepts are central to this proposal:

- **Concept drift** refers to the change over time in the conditional distribution of the process generating the data. Let $\{\mathbf{x}_t\}_{t=1}^{T}$ be a sequence of observations with $\mathbf{x}_t \in \mathbb{R}^{d_x}$. Concept drift is defined as:
$$p(\mathbf{x}_{t_1+1:t_1+F} \mid \mathbf{x}_{t_1-H+1:t_1}) \neq p(\mathbf{x}_{t_2+1:t_2+F} \mid \mathbf{x}_{t_2-H+1:t_2})$$

for distinct $t_1, t_2$, where $H$ is the historical length and $F$ is the forecasting length.

- **Non-stationarity** is the broader phenomenon of time-varying statistics, encompassing trend, seasonality, and variance changes (as well as concept drift)
- **Test-time adaptation (TTA)** is an emerging paradigm, has the potential to adapt a pre-trained model to data during testing, before making predictions.
- **Parameter-efficient fine-tuning (PEFT)**, specifically low-rank adaptation (LoRA[4]), updates only a small set of trainable parameters while freezing the base model, enabling rapid adaptation without catastrophic forgetting. For a pre-trained weight matrix $\mathbf{W}_0 \in \mathbb{R}^{d_h \times d_h}$, LoRA parametrizes the update as $\mathbf{W} = \mathbf{W}_0 + \mathbf{BA}$, where $\mathbf{B} \in \mathbb{R}^{d_h \times r}$, $\mathbf{A} \in \mathbb{R}^{r \times d_h}$, $r \ll d_h$, and $\mathbf{h} = \mathbf{Wx} = \mathbf{W}_0\mathbf{x} + \mathbf{BAx}$.

## 4 Related Works

### 4.1 Deep Learning-Based Time Series Forecasting

**Traditional Architectures.** Early deep learning for time series was dominated by Recurrent Neural Networks (RNNs) and Temporal Convolutional Networks (TCNs). While RNNs leverage gated units to model temporal dependencies, they are computationally inefficient and compress history into hidden states that become outdated under concept drift. TCNs enable parallelization but assume local stationarity, making them ill-suited for the shifting distributions of real-world data. Recently, MLP-based models like DLinear [16], N-BEATS [10], and TimeMixer [12] have emerged as efficient alternatives. However, these architectures fundamentally lack mechanisms for dynamic weight adaptation or variable interaction, leading to catastrophic forgetting during online updates.

**Transformers.** Transformers capture long-range dependencies through the self-attention mechanism:

$$\text{Attention}(\mathbf{Q}, \mathbf{K}, \mathbf{V}) = \sigma\left(\mathbf{Q}\mathbf{K}^\top\right)\mathbf{V}.$$

This allows models such iTransformer [8], Autoformer [14], PatchTST [9], and TimeXer [13] to outperform recurrent or convolutional constraints. Crucially, the attention score matrix $\mathbf{Q}\mathbf{K}^\top$ provides an inspectable representation of temporal dependencies. Unlike RNNs or MLPs, these attention maps allow Transformers to serve as a diagnostic tool for detecting and characterizing concept drift in real time.

### 4.2 Time Series Test-Time Adaptation

Test-time adaptation (TTA) enables models to adapt to out-of-distribution data in real time.

**Passive Adaptation.** Passive adaptation methods, such as TAFAS [5], PROCEED [20], and LEAF [22], rely on delayed ground-truth labels for supervised model updates. These methods incur high computational overhead and fail in environments where labels are unavailable, such as privacy-constrained healthcare or high-frequency financial markets.

**Active Adaptation.** Active adaptation methods like CEP [17] update the model only upon drift detection. However, existing active frameworks often rely on hand-crafted statistics that fail to capture complex multivariate shifts and face scalability issues as the number of detected regimes grows.

### 4.3 Attention-Based Concept Drift Handling

Recent works have integrated attention into drift pipelines. TAPN-CDD [6] uses temporal attention for error classification, while DSA-AE [1] evaluates reconstruction errors via dual attention blocks. Similarly, GLVTI [15] utilizes graph attention to capture evolving non-Euclidean correlations. Despite these advancements, attention is typically used as a peripheral component for detection or feature extraction. Current literature lacks a unified framework that jointly leverages attention maps for both unsupervised drift detection and real-time, parameter-efficient model adaptation.

## 5 Challenges

The primary challenge of this project lies in data. First, real-life time series datasets provide no ground-truth labels for concept drifts, which make the task difficult to quantitatively evaluate drift detection accuracy. Second, synthetic drift generators offer ground-truth labels but they rarely reflect the complex, high-dimensional, and non-stationary behaviors of real-world time series. Third, dataset construction strategies that combine distinct scenarios of sensor data provide easily labeled change points; however, it ignores the transition dynamics that characterize real concept drift.

## 6 Objectives

The primary objective of this project is to build a unified framework that handles concept drift in time series forecasting by coupling drift detection directly to an adaptation mechanism, forming a closed-loop system capable of maintaining forecasting accuracy across non-stationary data regimes.

The first objective is to develop an efficient concept drift detector by analyzing attention maps in transformer-based forecasting models. Specifically, the project will leverage pre-softmax attention dynamics to capture directional shifts in temporal and channel dependencies. Three models: iTransformer [8], TimesXer [13], and PatchTST [9] will be replicated, and various distance and similarity functions will be compared across drift types and severity levels using accuracy, precision, recall, F1-score, and detection latency on USC-HAD [18] and CaDrift [2]. The best-performing combination will then be benchmarked against DSA-AE [1], FEDD [3], and CEP [17].

The second objective is to enable lightweight test-time adaptation through parameter-efficient fine-tuning (PEFT), specifically Low-Rank Adaptation (LoRA) [4]. The adaptation module will be evaluated on Weather, Traffic, ETTs, and Exchange datasets using MAE and MSE, with performance compared against a frozen baseline and a full fine-tuning upper bound across all three transformer backbones, as well as against adaptation methods CEP [17], TAFAS [5], Proceed [20], and LEAF [22].

# 7 Proposed Methodology

## 7.1 Data Preprocessing

The dataset is split into training, validation, and testing sets. The training set is normalized, and the mean and variance statistics computed from the training set are saved to normalize the validation and testing sets. Each training sample consists of a historical input sliding window of length $H$ and a future output sliding window of length $F$ for supervised forecasting. Only the training set is randomly shuffled to break temporal correlations between each batch during training. For the validation and testing sets, samples are generated as non-overlapping sliding windows separated by the prediction length $F$, ensuring the output windows do not overlap and that evaluation respects temporal ordering.

## 7.2 Training Phase

The training phase consists of two stage: a pre-training stage and a concept discovery stage.

**Pre-Training Stage**. A transformer-based forecaster is pre-trained on the training set using standard supervised forecasting loss (e.g., MAE and MSE) and hyperparameters are tuned on the validation set. The forecaster's self-attention mechanism produces attention maps $\mathbf{S}_t \in \mathbb{R}^{n_{\text{token}} \times n_{\text{token}}}$ for each layer and head, where $n_{\text{token}}$ is the number of input tokens. These attention maps serve as structured representations of the temporal dependencies learned by the model.

**Concept Discovery Stage**. After pre-training, concept clusters are discovered using the validation set. For each validation sample, the forward pass is performed without gradient updates, and the attention maps from all layers and heads are extracted, smoothed, and aggregated into a single attention map representation per sample. Hierarchical clustering is then applied to these attention map representations using a distance metric such as Jensen-Shannon divergence, Wasserstein distance, or Frobenius norm. The resulting clusters correspond to distinct temporal dependency patterns, i.e., different concepts. For each cluster, a LoRA module is initialized. Each LoRA module is fine-tuned on the validation samples assigned to its corresponding concept while keeping the base forecaster frozen. This stage produces a concept cluster store containing cluster centroids $\mathbf{C}_c$ and their associated LoRA parameters $\mathbf{A}_c, \mathbf{B}_c$.

## 7.3 Testing Phase

The inference pipeline is illustrated in Figure 1 and comprises four steps: attention map extraction, concept drift detection, cluster retrieval, and model adaptation.

**Attention Map Extraction**. During each forward pass of the test sample, the forecaster generates an attention map, which is smoothed with the previous attention maps, aggregated across attention heads and layers, and compared against the currently active cluster centroid.

**Concept Drift Detection**. A concept drift is detected when the similarity between the current attention map and the active cluster centroid falls below a predefined threshold (or when the distance exceeds a threshold). This threshold can be calibrated using validation set statistics.

**Cluster Retrieval**. Upon drift detection, the cluster store retrieves the centroid most similar to the current attention map. The corresponding LoRA module associated with that cluster is also retrieved for model adaptation.

**Model Adaptation**. The retrieved LoRA module is applied to the forecaster, updating its parameters in a parameter-efficient manner. The forecaster then continues predictions using the adapted model until the next drift is detected.

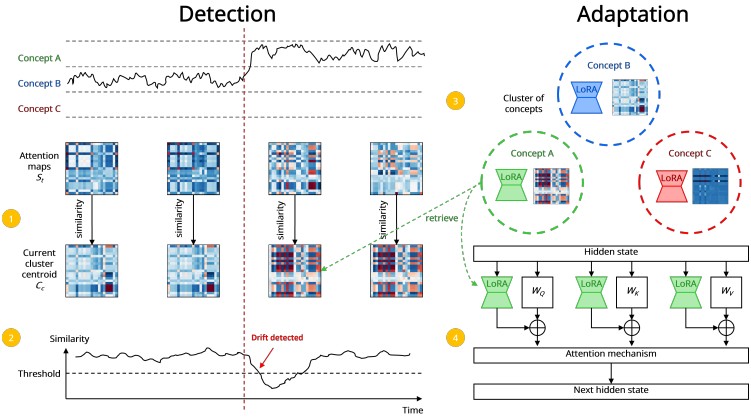

Figure 1: Inference pipeline of the proposed framework.

## 8 Dataset

We evaluate our framework on two drift-specific datasets and four long-horizon forecasting benchmarks.

**Drift Detection Benchmarks.** The **USC-HAD** [18] dataset provides 12 human activities from 14 subjects, captured via 3-axis inertial sensors. We treat transitions between activity classes as ground-truth drift points. **CaDrift** [2] is used to generate synthetic streams via Structural Causal Models (SCMs), allowing for controlled abrupt, incremental, and gradual shifts across distributional and covariate types. For both, preprocessing involves window segmentation and channel-wise normalization.

**Adaptation Benchmarks.** We utilize four standard unlabeled datasets to evaluate TTA performance: **ETT** (electricity transformer temperature), **Weather** (21 meteorological indicators), **Traffic** (hourly road occupancy), and **Exchange** (daily currency rates). These represent diverse regimes of seasonality and non-stationarity. Following standard protocols, all datasets undergo z-score normalization and sliding window segmentation ($H \in \{96, 720\}$; $F \in \{96, 192, 336, 720\}$).

## 9 Project Schedule

| Week | Dates | Task | Details | Lead/Support |
|------|-------|------|---------|--------------|
| 1–2 | Apr 27 – May 7 | **Backbone Replication** | Replicate TimeXer and PatchTST. | Alina, Elvin |
| 1–2 | Apr 27 – May 7 | **Experiment on detection techniques** | Try different detection techniques on different transformer models. | Ekkanat |
| 3–4 | May 8 – May 21 | **Detection Baseline Replication + Comparison** | Replicate existing solutions. | Alina, Azza, Elvin |
| 3–4 | May 8 – May 21 | **Model adaptation** | Apply LoRA and fine-tune models. | Ekkanat |
| 5–6 | May 22 – Jun 2 | **Adaptation Baseline Replication + Comparison** | Replicate existing solutions. | Alina, Azza, Elvin |
| 5–6 | May 22 – Jun 2 | **Final Report** | Results synthesis, documentation. | Full Team |

## 10 Progress So Far

**Completed:** The team has completed a comprehensive literature review focusing on time-series forecasting backbones, specifically TimeXer [13] and PatchTST [9], alongside state-of-the-art drift/concept detection and adaptation frameworks such as DSA-AE [1], FEDD [3], and CEP [17]. Furthermore, the iTransformer [8] backbone has been fully implemented. In terms of data preparation, the USC-HAD [18] and weather datasets have been preprocessed, and synthetic data has been successfully generated using the CaDrift [2] generator.

**Ongoing:** Current work is primarily focused on replicating existing backbones and baselines, as well as designing several experiments to identify the most effective detection techniques. Also, we are initiating three foundational empirical studies to demonstrate that attention maps are capable of capturing concept, data, and label drift; that attention maps from the same concept exhibit similar patterns; and that concepts occurring in the training phase will recur during the testing phase. Following the conclusion of these experiments, the project will move into the implementation of the adaptation module.

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
